# A Pilot Study on the Effects of a 10-Session Underwater Treadmill Programme on Canine Joint Range of Motion

**DOI:** 10.3390/ani15213186

**Published:** 2025-11-01

**Authors:** Julia Twarowska, Janusz Strychalski, Andrzej Gugołek

**Affiliations:** Department of Fur-Bearing Animal Breeding and Game Management, University of Warmia and Mazury in Olsztyn, Oczapowskiego 2, 10-719 Olsztyn, Poland; khzfil@uwm.edu.pl (J.T.); gugolek@uwm.edu.pl (A.G.)

**Keywords:** dogs, underwater treadmill, range of motion, joint mobility, physical rehabilitation

## Abstract

**Simple Summary:**

Dogs affected by musculoskeletal and neurological disorders often struggle with stiffness and limited mobility, which can reduce their quality of life. Rehabilitation aims to restore movement and comfort, and one method that is increasingly used is the underwater treadmill (UWTM). Exercising in water reduces the impact on joints while encouraging active movement and muscle strengthening. In this study, we reviewed the records of 50 dogs that participated in a programme consisting of ten UWTM sessions over five weeks. Before and after the programme, we measured the range of motion in the main limb joints using a goniometer. The results showed measurable improvements in both flexion and extension angles, indicating that overall joint mobility increased. These improvements were observed in dogs with different conditions and were not influenced by the dogs’ age, suggesting that the therapy can be useful for a broad range of patients. The findings highlight the potential benefits of repeated UWTM sessions as part of multimodal rehabilitation programmes. This practical and non-invasive method may therefore play an important role in helping dogs regain mobility and improve daily functioning.

**Abstract:**

Underwater treadmill (UWTM) therapy is increasingly applied in canine rehabilitation, yet evidence on its effects after multiple sessions on joint mobility remains limited. The aim of this pilot study was to evaluate the impact of a 10-session UWTM programme on passive range of motion (PROM) in dogs with various disorders. Clinical records from 50 dogs were analysed. Each patient completed two 20 min sessions per week over five consecutive weeks. PROM in the carpal, elbow, shoulder, tarsal, stifle, and hip joints was measured using a goniometer before and after the programme. After ten sessions, a significant improvement was observed in all joints, both in flexion and extension. Flexion angles decreased from 2.89% in the tarsal joint to 12.21% in the carpal joint, while extension angles increased from 0.61% in the elbow to 2.55% in the stifle joint. Consequently, overall PROM improved, with median increases ranging from 1.9% in the tarsus to 5.6% in the hip. These improvements were observed consistently across diagnostic groups. No significant correlations were found between age and the degree of PROM improvement. In summary, the findings indicate that a 10-session UWTM programme is associated with measurable improvements in joint mobility and may be a valuable component of multimodal canine rehabilitation.

## 1. Introduction

Underwater treadmill (UWTM) exercise is increasingly applied as a therapeutic method in veterinary medicine, where it is used for the management of musculoskeletal and neurological disorders, as well as for conditioning in various animal species, including horses [1], cats [2,3], and particularly dogs [4,5]. The unique physical properties of water, particularly buoyancy, viscosity, and resistance, provide therapeutic advantages that cannot be achieved during land-based exercise. Buoyancy reduces loading on joints and limbs, enabling safe activity in patients with pain or degenerative joint disease, while viscosity and resistance increase muscular effort, stimulate fibre activation, and enhance endurance. Furthermore, the slowing and magnifying effects of water on movement facilitate gait retraining and improve motor coordination [6].

It has been demonstrated that even a single UWTM session can significantly increase joint range of motion (ROM) and stride length in dogs [7]. Hydrotherapy also supports the management of osteoarthritis by reducing symptom severity and decreasing the need for analgesics [8]. The use of a treadmill platform enables dogs with hindlimb paresis to support themselves on the forelimbs while simultaneously exercising the hindlimbs [6]. Kinematic analyses have shown that UWTM walking, particularly when performed with an incline, can influence spinal segment motion (T1, T13, L7), which is clinically relevant for patients with spinal disorders [9]. Moreover, hydrotherapy is valuable in the rehabilitation of neurological patients, such as dogs with intervertebral disc disease, and can be incorporated into weight-reduction programmes, improving both body condition and cardiorespiratory fitness in obese dogs [6].

Measurement of joint passive range of motion (PROM) using a universal plastic goniometer is considered a reliable and repeatable method of joint assessment [10,11,12]. The effectiveness of UWTM therapy has also been evaluated using kinematic analysis based on reflective markers placed on anatomical reference points, as well as video recordings, which enable assessment of parameters such as ROM, stride length and stride frequency [7,13]. Additionally, acoustic myography (AMG) has been used to assess muscle activity, providing information on muscle-fibre recruitment [5].

Despite the promising results of studies investigating the application of underwater treadmill therapy in canine rehabilitation, there remains a need for further analyses to better understand the mechanisms of action, to optimise treatment protocols, and to assess the long-term efficacy of this method [14]. A review of the available literature on the use of UWTM therapy suggests wide potential applications in the management of various canine conditions. Orthopaedic and neurological disorders such as hip and elbow dysplasia, intervertebral disc disease, spondylosis, and cauda equina syndrome are among the most frequent causes of mobility impairment in dogs, accounting for a substantial proportion of rehabilitation cases in clinical practice [15,16]. However, studies investigating the effects of more than two UWTM sessions on joint ROM in dogs are scarce, with previous research limited mainly to one or two sessions [4,7,9,13]. Therefore, the aim of the present study was to determine the effect of a 10-session underwater treadmill programme on PROM in the fore- and hindlimb joints of dogs with diverse disorders. Furthermore, it was assessed whether therapeutic efficacy was influenced by the age of the animals.

## 2. Materials and Methods

Clinical records of 50 dogs that participated in a 10-session underwater treadmill (UWTM) programme at the KanVet clinic in Olsztyn, Poland, were analysed. The study population consisted of 28 males (10 neutered) and 22 females (12 spayed), with a mean age of 6.26 ± 3.03 years (range 1–12 years). The dogs represented various breeds and body weights (Table 1) and were referred for UWTM therapy owing to different musculoskeletal or neurological disorders (Table 2). All patients had completed standard medical or surgical treatment and were considered clinically stable at the time of enrolment. Disorders included orthopaedic and neurological conditions typically encountered in rehabilitation practice, such as intervertebral disc disease, hip dysplasia, spondylosis, cauda equina syndrome, patellar luxation, elbow dysplasia, and postoperative recovery following femoral fracture repair. In each case, hydrotherapy was initiated after the exclusion of acute inflammation, pain, or other contraindications to underwater treadmill exercise.

A WATER-WALKER^®^ underwater treadmill (Ludwig KEIPER GmbH & Co. KG, Obermoschel, Germany), with an internal chamber of approximately 1.75 m (length) × 0.79 m (width) × 1.20 m (height) was used for rehabilitation (Figure 1). All UWTM sessions were conducted by the same certified canine rehabilitation therapist. The UWTM programme consisted of 20 min sessions (excluding treadmill familiarisation), conducted twice weekly over five consecutive weeks. The water level was set at approximately three-quarters of the femur length, as this depth provided optimal buoyancy and resistance during underwater treadmill exercise. At this immersion level, body-weight support is substantially reduced (by about 60–70%), thereby minimising joint loading while maintaining sufficient resistance to promote muscular activity and joint mobility [17]. A treadmill speed of 1.2 m/s (4.32 km/h) was used as the primary setting, with minor adjustments made for individual dogs ranging from 0.9 to 1.3 m/s to maintain a natural walking gait depending on their size, comfort, and clinical condition. The water temperature was maintained between 26 and 30 °C. All parameters described above fell within the ranges recommended for UWTM hydrotherapy in standard veterinary rehabilitation guidelines [6,17].

All dogs included in the study did not receive any additional physiotherapeutic or pharmacological treatments during the 10-session UWTM programme. The enrolled patients had stable chronic orthopaedic or neurological conditions, with no acute exacerbations during the study period.

Informed consent was obtained from each dog owner. Passive joint angles of extension and flexion were measured using a transparent, universal 12-inch (30 cm), 360° plastic goniometer (Baseline^®^; Fabrication Enterprises Inc., White Plains, NY, USA), following the procedures described by Reusing et al. [12]. Measurements were performed with the dog positioned in lateral recumbency on both left and right sides. For each dog, PROM was measured in the carpal, elbow, shoulder, tarsal, stifle, and hip joints before and after completion of the 10-session UWTM programme. Each angle was measured three times on both sides of the body, and mean values were calculated for each joint. All measurements were performed by the same examiner to ensure consistency.

PROM values before and after the 10-session UWTM programme were analysed using a repeated-measures ANOVA, with time (pre vs. post) as the within-subject factor and sex, breed, and disorder as between-subject factors. The general linear model was specified as:Y_ijkl_ = μ + α_i_ + β_j_ + γ_k_ + δ_l_ + α_i_β_j_ + α_i_γ_k_ + β_j_γ_k_ + α_i_β_j_γ_k_ + α_i_δ_l_ + β_j_δ_l_ + γ_k_δ_l_ + ε_ijkl_,
where µ is the general mean, α_i_ is the effect of time, β_j_ is the effect of sex, γ_k_ is the effect of breed, δ_l_ is the effect of disorder, interaction terms represent the combined effects of these factors, and ε_ijkl_ is the random error.

In analyses conducted within individual disorders, PROM values before and after therapy were compared using repeated measures ANOVA models, including sex and breed as between-subject factors, but excluding the disorder effect.

Correlations between age and percentage improvement in PROM were assessed using Spearman’s rank correlation coefficient (rho). All analyses were performed using R software (v.4.5.1) [18].

## 3. Results

The 10-session UWTM programme resulted in a reduction in flexion angles in all evaluated joints—carpal, elbow, shoulder, tarsal, stifle, and hip (Table 3; *p* < 0.001 in all cases). The decrease in flexion angle ranged from 2.89% in the tarsal joint to 12.21% in the carpal joint. Importantly, the reduction in flexion angle was significant across all examined conditions (intervertebral disc disease, hip dysplasia, spondylosis, and others; *p* < 0.001 in each case and overall).

Completion of the 10-sessionUWTM programme increased extension angles in all evaluated joints (Table 4). Statistically significant differences were observed in all joints assessed (*p* < 0.001) and across all disorder groups (*p* < 0.001 in all cases and overall). The increase in extension angles ranged from 0.61% in the elbow joint to 2.55% in the stifle joint.

As shown in Figure 2, completion of the 10-session UWTM programme increased PROM values in all joints (*p* < 0.001 in all cases). The median PROM for the carpus increased from 165.5° before therapy to 171.5° after therapy (improvement of 3.61%), for the elbow from 137.0° to 140.7° (2.98%), for the shoulder from 119.5° to 124.2° (2.82%), for the tarsus from 128.2° to 131.0° (1.95%), for the stifle from 116.5° to 122.5° (5.20%), and for the hip from 107.5° to 113.5° (5.62%).

Spearman’s rank correlation coefficients between the dogs’ age and the percentage improvement in PROM were as follows: carpus, rho = −0.059; elbow, rho = 0.198; shoulder, rho = −0.064; tarsus, rho = −0.023; stifle, rho = −0.044; and hip, rho = 0.119. In all cases, the correlations were not statistically significant (*p* > 0.05), suggesting no meaningful relationship between the dogs’ age and therapeutic response (Figure 3).

## 4. Discussion

Despite the increasing use of UWTM therapy in dogs, relatively few studies have evaluated its effect on joint ROM in this species [19,20]. Moreover, to date, research on the longer-term impact of UWTM therapy on ROM in dogs remains extremely limited [21]. Barnicoat and Wills [4], for example, assessed ten healthy Labrador Retrievers at four immersion depths and reported increases in stride length of up to 14% and decreases in stride frequency of up to 9% after two sessions at each depth. Preston and Wills [7] evaluated twelve Labradors with elbow dysplasia and found that a single UWTM session increased elbow ROM from 38.9° to 43.1° and lengthened stride by approximately 5%. Bliss et al. [13] demonstrated that immersion depth strongly influences limb kinematics, with stifle flexion increasing by up to 15° at hock-level water depth in ten dogs, while Hodgson et al. [9] showed in eight dogs that spinal kinematics (T1-L7) were not significantly altered across five UWTM conditions, indicating safety but limited functional change. To the best of our knowledge, the present study is the first to evaluate a 10-session UWTM programme in a larger and more clinically diverse cohort (*n* = 50). We observed consistent improvements across all joints, with flexion angles reduced by 2.9–12.2% and extension angles increased by 0.6–5.6%. These findings correspond to prior reports in dogs where a single hydrotherapy session increased joint range of motion in cases of elbow dysplasia [7] and in combined hydrotherapy-laser interventions in dogs with hip dysplasia [21]. However, no control group was included; therefore, the study should be regarded as a preliminary pilot study. The absence of a control group was due to the retrospective use of clinical rehabilitation records, where withholding therapy or assigning sham treatments would not have been ethically acceptable [8,22,23]. It should also be noted that Crook et al. [8] similarly evaluated changes in joint mobility in dogs without including a control group, following a comparable methodological approach. A dry treadmill cannot be considered a control in this context, as dogs are typically referred to dry treadmill exercise at different stages of rehabilitation and with different clinical profiles (e.g., sport conditioning or late-phase recovery) compared to those referred to UWTM therapy, where buoyancy is essential for managing pain, obesity, or neurological impairment [13,17].

In dogs diagnosed with hip dysplasia, the pattern of changes in joint mobility observed in our study closely resembles that reported by Reusing et al. [21], who evaluated twelve hydrotherapy sessions in a comparable clinical population. In our cohort (*n* = 12), hip flexion decreased from 47.7° to 45.1° (2.6°, 5.4%), while extension increased from 155.6° to 159.1° (3.5°, 2.3%), resulting in a mean PROM increase of 6.1° (5.6%). In the hydrotherapy-only group described in the aforementioned study, mean flexion changed slightly from 59.2° to 58.1° (1.1°, 1.8%), whereas extension increased from 129.0° to 137.2° (8.2°, 6.4%), corresponding to an overall PROM rise of 9.4° (7.5%). Despite the longer duration of their programme (12 vs. 10 sessions), the relative percentage improvements were comparable, confirming that repeated aquatic sessions promote measurable increases in hip joint mobility in dogs with orthopaedic disorders. Our study, however, is broader, and its results demonstrate that 10 UWTM sessions were associated with reductions in flexion angles and increases in extension angles across all evaluated joints, leading to an overall improvement in PROM. Although the magnitude of these changes was modest, previous research in canine rehabilitation has demonstrated that even small improvements in joint mobility can be accompanied by measurable functional gains, including an accelerated recovery of limb use, decreased lameness, enhanced joint functionality, and more symmetrical gait parameters, leading to a gradual return to normal load distribution in the affected limb [11,24,25,26]. Nevertheless, while statistically significant, the present results should be interpreted with caution. This is a limitation of the study, and further research including functional and quality-of-life outcomes, as well as pain assessment tools such as the Helsinki Chronic Pain Index and the Canine Brief Pain Inventory, would be valuable to clarify their practical significance. That said, the changes observed in our study translated into improved mobility, which is of clear benefit to canine patients undergoing rehabilitation. These improvements were observed consistently across all defined groups of dogs. These results therefore provide support for the inclusion of UWTM therapy as a standard component of multimodal canine rehabilitation.

This is also the first study to examine whether the efficacy of UWTM therapy is influenced by the age of the dogs. Previous publications did not include age as a variable. Barnicoat et al. [4] and Bliss et al. [13] evaluated the effects of UWTM sessions at varying water depths but did not analyse age-dependent responses. Preston et al. [7] and Hodgson et al. [9] assessed short-term changes in ROM after a single session without stratifying results by age. Although ageing in dogs is commonly associated with reduced joint flexibility [27,28], our study found no significant correlation between the dogs’ age and the percentage improvement in PROM following the 10-session UWTM programme. This finding is clinically reassuring, as it suggests that UWTM therapy can be safely and effectively implemented in canine patients of all ages.

In our study, dogs immersed to approximately three-quarters of the femoral length showed increased PROM (Table 3 and Table 4). These findings are in line with the observations of Vitger et al. [5], who showed that immersion depth influences activation of the biceps femoris and vastus lateralis muscles in dogs, with the highest activity at mid-femur immersion—closely corresponding to the level we adopted. Immersion to the hip joint has been reported to reduce body-weight bearing by 62% [16], while immersion above the hip joint decreases activation of the gluteus medius and longissimus dorsi compared with lower levels [29]. Notably, Mendez-Angulo et al. [30] demonstrated in horses that UWTM exercise at different water depths influences joint ROM in a depth-dependent manner. For example, the greatest increases in fetlock ROM were observed at immersion to the fetlock or tarsal joint, while the greatest increases in carpal ROM occurred at immersion to the tarsal joint. Immersion to the stifle joint improved ROM across the fetlock, carpal, and tarsal joints, suggesting a multi-joint effect of deeper immersion. Beyond these biomechanical mechanisms, it is also plausible that the improvement observed across several joints in our study was influenced by broader physiological adaptations associated with aquatic exercise. Fernandes et al. [31] showed that underwater treadmill sessions in healthy dogs promote a slower and deeper respiratory pattern without elevating blood lactate or heart rate, confirming the aerobic nature and safety of this training modality. In turn, Lewis et al. [32] adapted a land-based gait scoring system for underwater treadmill use in dogs recovering from thoracolumbar intervertebral disc extrusion and found that underwater locomotion facilitated earlier detection of coordinated motor activity and supported progressive gait recovery even in non-ambulatory patients. Taken together, these findings suggest that aquatic exercise integrates biomechanical unloading with systemic conditioning and neuromotor stimulation, providing a comprehensive mechanism underlying the multijoint improvement observed in our study. Further research should refine hydrotherapy parameters such as immersion depth, session frequency, and workload to optimise both locomotor restoration and overall physiological adaptation in dogs with different pathologies.

## 5. Conclusions

A 10-session UWTM programme was associated with improvements in both flexion and extension angles in all evaluated joints of the thoracic and pelvic limbs in dogs. These effects were observed across the defined diagnostic groups, including the group combining several less common disorders, and did not appear to depend on the dogs’ age.

## Figures and Tables

**Figure 1 animals-15-03186-f001:**
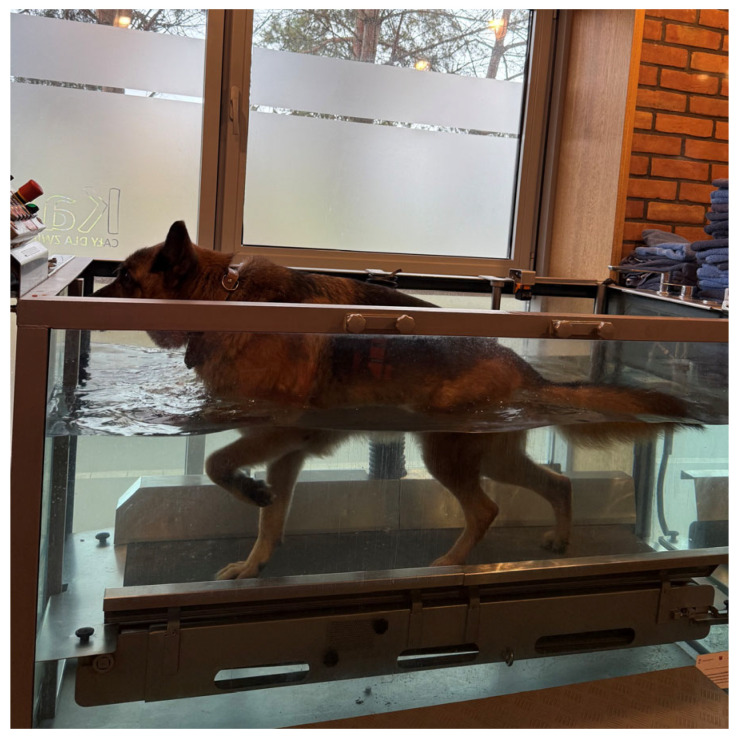
Dog walking on an underwater treadmill.

**Figure 2 animals-15-03186-f002:**
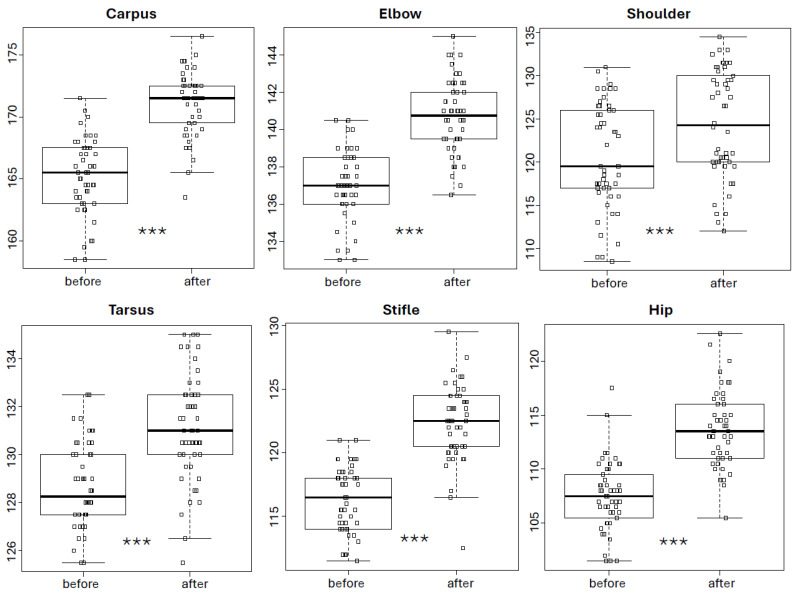
Passive range of motion (PROM) of joints in dogs before and after the 10-session underwater treadmill (UWTM) programme (medians, 25th and 75th percentiles, adjacent values, and outliers). *** indicates a significant difference between mean values before and after the underwater treadmill (UWTM) programme (*p* < 0.001).

**Figure 3 animals-15-03186-f003:**
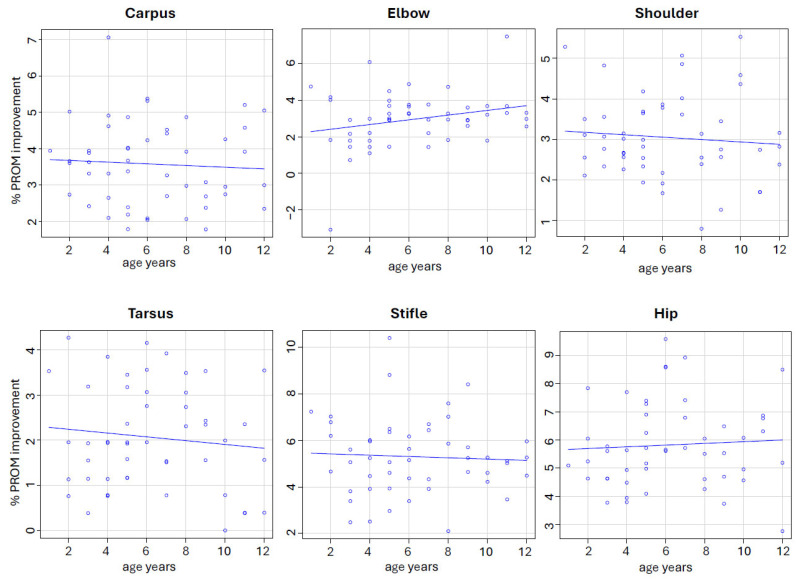
Correlations between age and percentage improvement in passive range of motion (PROM) in dogs. None of the correlations were statistically significant (*p* > 0.05).

**Table 1 animals-15-03186-t001:** Dog breeds and body weights of the study population (mean ± SD).

Breed	Dogs (*n*)	Body Weight (kg)	Breed	Dogs (*n*)	Body Weight (kg)
French Bulldog	9	12.4 ± 1.8	American Akita	1	47.0
Mixed-breed	9	18.6 ± 6.7	Beagle	1	13.2
German Shepherd	5	32.8 ± 3.5	American Staffordshire Terrier	1	26.4
Golden Retriever	4	31.2 ± 2.9	Pug	1	8.9
Jack Russell Terrier	3	6.9 ± 0.8	Cavalier King Charles Spaniel	1	7.3
White Swiss Shepherd	2	32.5 ± 2.1	Maltipoo	1	5.1
Labrador Retriever	2	32.7 ± 3.0	Miniature Schnauzer	1	8.2
Maltese	2	4.6 ± 0.5	Basenji	1	10.4
Cane Corso	2	44.2 ± 3.8	Yorkshire Terrier	1	3.6
Dachshund	2	8.2 ± 1.1	West Highland White Terrier	1	8.1

**Table 2 animals-15-03186-t002:** Types and clinical conditions of disorders in the study population.

Disorder	Cases (*n*)	Disorder Grade/Clinical Condition
Intervertebral disc disease	20	Completed pharmacological treatment and achieved neurological stability before inclusion
Hip dysplasia	12	FCI grades B–D
Spondylosis	8	chronic stiffness but no acute pain or severe lameness
Cauda equina syndrome	4	stable mild or moderate symptoms
Patellar luxation	3	Grades I–II according to Putnam’s scale
Elbow dysplasia	2	chronic stiffness but no acute pain or severe lameness
Femoral fracture	1	completed postoperative recovery following surgical fixation

**Table 3 animals-15-03186-t003:** Flexion angles (°) of selected joints before and after 10-session underwater treadmill (UWTM) programme in dogs with various disorders (mean ± SD).

Joint	10 UWTM Sessions	Disorder				
		Intervertebral Disc Disease *n* = 20	Hip Dysplasia *n* = 12	Spondylosis *n* = 8	Others *n* = 10	Total
Carpus	before	23.8 ± 3.2	23.9 ± 2.2	24.6 ± 2.7	25.0 ± 2.4	24.2 ± 2.7
	after	20.7 ± 3.4 ***	21.1 ± 2.7 ***	21.9 ± 2.9 ***	22.2 ± 2.6 ***	21.3 ± 3.0 ***
Elbow	before	24.7 ± 3.3	23.5 ± 2.9	25.9 ± 3.6	23.4 ± 2.5	24.3 ± 3.2
	after	22.7 ± 3.4 ***	21.8 ± 2.9 ***	23.6 ± 3.2 ***	21.2 ± 2.5 ***	22.3 ± 3.1 ***
Shoulder	before	40.9 ± 7.2	40.5 ± 6.1	39.0 ± 5.7	39.7 ± 7.6	40.3 ± 6.6
	after	39.0 ± 7.5 ***	38.8 ± 6.3 ***	37.3 ± 5.9 ***	37.9 ± 7.6 ***	38.5 ± 6.8 ***
Tarsus	before	38.9 ± 1.4	37.8 ± 2.4	38.5 ± 2.9	37.9 ± 1.2	38.4 ± 2.0
	after	37.8 ± 1.7 ***	36.8 ± 2.7 ***	37.6 ± 3.4 ***	36.5 ± 1.3 ***	37.3 ± 2.2 ***
Stifle	before	42.7 ± 2.7	43.0 ± 1.9	42.1 ± 3.6	42.8 ± 2.2	42.7 ± 2.5
	after	39.3 ± 2.9 ***	39.6 ± 2.2 ***	38.9 ± 3.4 ***	38.8 ± 2.0 ***	39.2 ± 2.6 ***
Hip	before	48.3 ± 3.8	47.7 ± 3.9	48.8 ± 3.3	50.0 ± 2.8	48.6 ± 3.6
	after	45.4 ± 3.9 ***	45.1 ± 4.0 ***	46.5 ± 3.8 ***	47.6 ± 2.6 ***	45.9 ± 3.7 ***

*** indicates a significant difference between mean values before and after the underwater treadmill (UWTM) programme (*p* < 0.001).

**Table 4 animals-15-03186-t004:** Extension angles (°) of selected joints before and after 10-session underwater treadmill (UWTM) programme in dogs with various disorders (mean ± SD).

Joint	10 UWTM Sessions	Disorder				
		Intervertebral Disc Disease *n* = 20	Hip Dysplasia *n* = 12	Spondylosis *n* = 8	Others *n* = 10	Total
Carpus	before	188.5 ± 5.1	189.4 ± 1.6	188.4 ± 3.7	190.9 ± 3.3	189.2 ± 3.9
	after	191.1 ± 4.8 ***	193.0 ± 1.6 ***	191.8 ± 3.8 ***	193.6 ± 2.7 ***	192.2 ± 3.7 ***
Elbow	before	161.4 ± 3.1	159.8 ± 2.8	161.3 ± 1.5	160.3 ± 3.0	160.8 ± 2.8
	after	163.2 ± 2.8 ***	162.3 ± 2.3 ***	163.8 ± 1.8 ***	161.8 ± 3.2 ***	162.8 ± 2.7 ***
Shoulder	before	160.8 ± 2.1	160.9 ± 1.7	161.9 ± 2.2	160.5 ± 2.0	161.0 ± 2.0
	after	162.8 ± 1.9 ***	163.1 ± 1.4 ***	163.4 ± 1.8 ***	162.1 ± 2.4 ***	162.8 ± 1.9 ***
Tarsus	before	167.4 ± 2.3	166.3 ± 2.6	165.6 ± 2.7	167.2 ± 1.4	166.8 ± 2.3
	after	168.9 ± 2.3 ***	168.4 ± 1.7 ***	166.8 ± 2.2 ***	168.4 ± 1.3 ***	168.3 ± 2.0 ***
Stifle	before	158.7 ± 3.3	159.7 ± 2.7	158.1 ± 3.2	158.6 ± 1.8	158.8 ± 2.9
	after	161.4 ± 3.0 ***	162.1 ± 3.2 ***	161.1 ± 3.3 ***	161.3 ± 2.4 ***	161.5 ± 2.9 ***
Hip	before	156.4 ± 3.1	155.6 ± 2.8	155.6 ± 2.7	156.5 ± 3.2	156.1 ± 2.9
	after	159.9 ± 3.5 ***	159.1 ± 3.5 ***	159.4 ± 2.4 ***	160.1 ± 3.4 ***	159.7 ± 3.3 ***

*** indicates a significant difference between mean values before and after the underwater treadmill (UWTM) programme (*p* < 0.001).

## Data Availability

The data presented in this study are available upon request from the corresponding author.

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
