# Peer review of "A Pilot Study on the Effects of a 10-Session Underwater Treadmill Programme on Canine Joint Range of Motion"

_animals, 2025, doi:10.3390/ani15213186_

Round 1

Reviewer 1 Report

Comments and Suggestions for Authors

The submitted manuscript concerns underwater treadmill therapy (UWTM) in
dogs and its impact on joint range of motion. The main aims of the study
were to assess whether 10 UWTM sessions improve joint mobility in dogs,
and to examine whether age influences the effectiveness of therapy. By
analysing clinical records from 50 dogs undergoing 10 sessions, the
authors provide novel data on multi-session therapy and its effects on
passive range of motion.

I consider the manuscript valuable. Although there is no control group
(due to its retrospective nature, which the authors explain in the
discussion), this approach is common both in veterinary and human
medicine. For example, the study by Crook et al. (2007), cited by the
authors, also measured joint mobility with a goniometer before and after
a programme and did not include a control group.

However, I suggest the following revisions:

In the abstract, specify that this is a pilot study (since no control
group was included).

In the last paragraph of the introduction, add information on how common
orthopedic and neurological conditions are in dogs (e.g., dysplasia,
spinal disorders).

In the Methods, clarify whether the sessions were always conducted by
the same therapist, or by several.

In the Discussion, add a note that Crook et al. (2007) also did not
include a control group (lines 180–187).

In the Discussion, it would be useful to compare the obtained results
with those of de Oliveira Reusing et al. (2021), where therapy lasted 12
sessions, although it concerned only the hip joint. At present, the
authors mention this publication only once, without referring to its
findings.

After addressing these comments, the manuscript will be better prepared.

Author Response

Dear Reviewer,

We thank the reviewer for the constructive and insightful comments, which have helped us improve the clarity and depth of our manuscript. Below, we provide our responses to your comments.

Comment 1:

In the abstract, specify that this is a pilot study (since no control group was included).

Response 1:

The term “pilot study” has been added to clarify that the research was exploratory and lacked a control group (line 27).

Comment 2:

In the last paragraph of the introduction, add information on how common orthopedic and neurological conditions are in dogs (e.g., dysplasia, spinal disorders).

Response 2:

We expanded the final paragraph to include data on the prevalence of orthopedic and neurological disorders in dogs, supported by appropriate references (Tomlinson 2012; Kongtawelert et al. 2014) (lines 79–83).

Comment 3:

In the Methods, clarify whether the sessions were always conducted by the same therapist, or by several.

Response 3:

We clarified that all UWTM sessions were performed by the same certified canine rehabilitation therapist (lines 120–121).

Comment 4:

In the Discussion, add a note that Crook et al. (2007) also did not include a control group (lines 180–187).

Response 4:

We added a note that Crook et al. (2007) also conducted their study without a control group, supporting our methodological rationale (lines 234–236).

Comment 5:

In the Discussion, it would be useful to compare the obtained results with those of de Oliveira Reusing et al. (2021), where therapy lasted 12 sessions, although it concerned only the hip joint. At present, the authors mention this publication only once, without referring to its findings.

Response 5:

We incorporated a comparison with Reusing et al. (2021), highlighting similar improvements in joint range of motion after repeated hydrotherapy sessions (lines 241–251).

Once again, we would like to thank you for taking the time to read our manuscript. We believe that these modifications fully address your comments and enhance the manuscript’s transparency and scientific context.

Sincerely,

The authors

Reviewer 2 Report

Comments and Suggestions for Authors

Review

45 – Rather than citing references related to humans, it would be preferable to reference the veterinary literature for this study.

72 – The order of the references appears to be inconsistent. Please revise to ensure they are listed sequentially, either by finding alternative citations or correcting the order.

76 – Similarly, in this sentence, is there a reason to mention humans? Since this paper concerns the application of UWTM to dogs, it would be sufficient and more appropriate to refer only to dogs.

85 – Please provide detailed information regarding the weights, precise sex (including neutered status), and breeds of the dogs participating in the study.

86 – Add an explanation regarding the water level applied to each animal patient, and include representative photographs if possible.

87 – Specify the brand name, manufacturer, and country of origin for the UWTM device used in this study.

89 – Clarify whether the listed conditions were treated with any other interventions and provide detailed information on disease progression. For patellar luxation cases, please indicate the grade.

94 – Is there a rationale for setting the speed at 1.2 m/s? Since walk and trot speeds vary depending on animal size, clarify whether this distinction was considered.

99 – Please include the brand name, manufacturer, and country of origin for the goniometer used.

126 – In the tables, please mark statistical significance with an asterisk (*) and provide explanatory notes below the table for clarity. The current presentation is not immediately intuitive.

142 – The same applies to figures. Instead of displaying p-values within the figures, it would be better to indicate significance only and insert appropriate notes below the figure.

181 – The main finding of this study is improvement associated with increased ROM. Instead of citing human references, please locate and discuss veterinary literature regarding ROM improvements, including at least two references, to enhance reader understanding of the study's utility.

190 – This is a limitation of the study. Further research using pain assessment tools such as the Helsinki chronic pain index and canine pain brief would be valuable.

222 – I recommend utilizing resources and literature from the veterinary field when discussing this study.

Author Response

Dear Reviewer,

We are grateful for all the valuable comments which were helpful in improving the manuscript. Due consideration was given to all suggestions. Below are our responses to the Reviewer’s comments.

Comment 1:

45 – Rather than citing references related to humans, it would be preferable to reference the veterinary literature for this study.

Response 1:

We have modified the first sentence to refer to animal studies instead of human research (lines 45–49).

Comment 2:

72 – The order of the references appears to be inconsistent. Please revise to ensure they are listed sequentially, either by finding alternative citations or correcting the order.

Response 2:

Thank you for this comment. We have checked the order of the citations, and it is generally correct. The citation numbers are not strictly sequential because they refer to publications cited earlier in the text.

Comment 3:

76 – Similarly, in this sentence, is there a reason to mention humans? Since this paper concerns the application of UWTM to dogs, it would be sufficient and more appropriate to refer only to dogs.

Response 3:

You are right that citing publications related to humans is not appropriate in this context. Therefore, we have removed this reference from the last paragraph of the Introduction.

Comment 4:

85 – Please provide detailed information regarding the weights, precise sex (including neutered status), and breeds of the dogs participating in the study.

Response 4:

Thank you for this comment. We have added this information (lines 91–101).

Comment 5:

86 – Add an explanation regarding the water level applied to each animal patient, and include representative photographs if possible.

Response 5:

The water level was set in the clinic at approximately three-quarters of the femur length. In the current version of the manuscript, we have added an explanation of this level (lines 125–128). We have also included a photograph of a dog on the underwater treadmill taken in this clinic (line 140).

Comment 6:

87 – Specify the brand name, manufacturer, and country of origin for the UWTM device used in this study.

Response 6:

We have added this required information (lines 118–120).

Comment 7:

89 – Clarify whether the listed conditions were treated with any other interventions and provide detailed information on disease progression. For patellar luxation cases, please indicate the grade.

Response 7:

Thank you for this valid comment. We have added the necessary information in lines 107–117 and 135–138.

Comment 8:

94 – Is there a rationale for setting the speed at 1.2 m/s? Since walk and trot speeds vary depending on animal size, clarify whether this distinction was considered.

Response 8:

This speed is consistent with standard veterinary rehabilitation guidelines [6,17]. In general, smaller dogs walk more slowly on a dry treadmill, but in water they experience much lower resistance than larger dogs, so their speed may be similar. However, you are right that the speed should sometimes be slightly adjusted. We have added this information (lines 129–131).

Comment 9:

99 – Please include the brand name, manufacturer, and country of origin for the goniometer used.

Response 9:

We have added this information (lines 144–145).

Comment 10:

126 – In the tables, please mark statistical significance with an asterisk (*) and provide explanatory notes below the table for clarity. The current presentation is not immediately intuitive.

Response 10:

Thank you for this helpful comment. We have simplified Tables 1 and 2.

Comment 11:

142 – The same applies to figures. Instead of displaying p-values within the figures, it would be better to indicate significance only and insert appropriate notes below the figure.

Response 11:

We have simplified Figures 2 and 3. As shown in Figure 3, there was no correlation between age and percentage improvement in PROM; therefore, we added a note about the absence of differences below this figure.

Comment 12:

181 – The main finding of this study is improvement associated with increased ROM. Instead of citing human references, please locate and discuss veterinary literature regarding ROM improvements, including at least two references, to enhance reader understanding of the study's utility.

Response 12:

We believe that in this case you are probably referring to the second paragraph of the Discussion. We have removed the references to human studies and instead expanded the discussion related to dogs.

Comment 13:

190 – This is a limitation of the study. Further research using pain assessment tools such as the Helsinki chronic pain index and canine pain brief would be valuable.

Response 13:

Thank you for this observation. We have expanded the limitations to include the information you suggested (lines 268–271).

Comment 14:

I recommend utilizing resources and literature from the veterinary field when discussing this study.

Response 14:

Thank you for this valid comment. We have made an effort to avoid references to human medicine. Only one such reference [31] remains in the entire text (lines 302–303).

Thank you again for devoting your time to reviewing our manuscript. We hope you find the improvements satisfactory.

Sincerely,

The authors

Round 2

Reviewer 2 Report

Comments and Suggestions for Authors

Dear Authors

Line 90-99: Rather than listing them, it's more valuable to make them visible by creating a table.

Line 102-111: The history of the problematic condition or participating dog should also be tabulated so that you can see it at a glance.

Line 122: If you adjusted the pace to suit the individual, please provide a range, and it would be scientifically helpful for researchers to clarify whether you want to walk or trot for UWTM exercises. I think this is a very important part of improving the ROM. Please make it more scientific

Table 1 : I would like to see the asterisks referenced to other literature, which is usually done in the top right corner of the results.

Line 279-281: The authors did a good job of finding and replacing the veterinary research citations instead of human research, but why are human studies mentioned again?

Author Response

Dear Reviewer,

We would like to thank the Reviewer for the time and effort devoted to evaluating our manuscript. The comments provided were very helpful in improving the scientific quality of the paper. We have addressed each point as detailed below.

Comment 1:

Line 90-99: Rather than listing them, it's more valuable to make them visible by creating a table.

Response 1:

Thank you for this helpful suggestion. We agree that presenting these data in a table provides a clearer and more concise overview. Accordingly, we have created a new Table 1 to visualise the information that was previously listed in the text (line 109).

Comment 2:

Line 102-111: The history of the problematic condition or participating dog should also be tabulated so that you can see it at a glance.

Response 2:

We have summarised the clinical history and main conditions of the dogs in a new Table 2, which allows these details to be viewed at a glance (line 111).

Comment 3:

Line 122: If you adjusted the pace to suit the individual, please provide a range, and it would be scientifically helpful for researchers to clarify whether you want to walk or trot for UWTM exercises. I think this is a very important part of improving the ROM. Please make it more scientific

Response 3:

As suggested, we have clarified in the revised text that the underwater treadmill speed was set at 1.2 m/s (4.32 km/h) as the primary pace, with minor adjustments ranging from 0.9 to 1.3 m/s to maintain a natural walking gait depending on the dogs’ size and clinical condition (lines 126-128).

Comment 4:

Table 1 : I would like to see the asterisks referenced to other literature, which is usually done in the top right corner of the results.

Response 4:

We appreciate the Reviewer’s suggestion. The asterisks indicating significant differences have now been added directly in the top right corner of the results. Below the tables, a description of these asterisks is provided. In the revised version of the manuscript, these are Tables 3 and 4.

Comment 5:

Line 279-281: The authors did a good job of finding and replacing the veterinary research citations instead of human research, but why are human studies mentioned again?

Response 5:

We thank the Reviewer for this careful observation. In the revised version, all human-related references have been removed. The sentence was rewritten to focus exclusively on veterinary research, citing Fernandes et al. (2022) and Lewis et al. (2023), who demonstrated the physiological and neuromotor effects of underwater treadmill exercise in dogs. The revised text now highlights canine-specific mechanisms, emphasising that the observed multijoint improvement likely reflects both biomechanical unloading and systemic adaptations associated with aquatic therapy.

We sincerely thank the Reviewer once again for the constructive and insightful feedback, which has been invaluable in improving the quality and clarity of our manuscript.

With kind regards,
The Authors